# IoT-Enabled Sustainable and Cost-Efficient Returnable Transport Management Strategies in Multimodal Transport Systems

**Yanqi Zhang** [1], **Xiaofei Kou** [1,*], **Haibin Liu** [1], **Shiqing Zhang** [1] **and Liangliang Qie** [2]

[1] School of Management Engineering, Zhengzhou University of Aeronautics, Zhengzhou 450046, China
[2] School of Chemistry and Material Science, Hubei Engineering University, Xiaogan 432000, China
* Correspondence: kxfei1987@zua.edu.cn

**Abstract:** Returnable transport items (RTIs) are widely used in multimodal transport systems. However, due to the lack of effective tracking methods, RTIs management efficiency is low and RTIs are easily lost, which directly and indirectly causes economic losses to enterprises. Internet of Things (IoT) technology is proved to be effective in realizing real-time tracking and tracing of various objects in diverse fields. However, an IoT-enabled RTIs management system in a multimodal transport system has not been widely accepted due to a lack of an effective cost decision model. To address these problems, this research first presents three typical schemes of RTIs management. through extensive field studies on collaborative logistics service providers in multimodal transport systems. Then, the cost–benefit analyses of these three schemes are conducted while the decision models on whether to adopt IoT technologies are built. Finally, based on the decision models, the main factors affecting the application of IoT-RTIs management systems are studied by numerical analysis, based on which several managerial implications are presented. These results can serve as a theoretical basis for enterprises interested in finding out whether IoT technology should be used in RTIs management.

**Keywords:** multimodal transport system; RTIs management; Internet of Things; cost–benefit analysis; decision models



## 1. Introduction

Multimodal transport refers to the transport of goods from the place of pickup over to the designated place of delivery by multimodal transport operators in at least two different modes of transport. The multimodal transport system is considered to be one of the important modes of sustainable transportation. Returnable transport items (RTIs), such as containers, pallets, crates, roll cages, work bins, and flat cars, are widely used in multimodal transport systems to reduce solid wastes in the logistics process, provide better protection for products, enable more efficient handling of materials, and finally facilitate green transport systems. The wide application of RTIs can reduce the waste of resources, actively promote carbon emission reduction, and promote the sustainable development of multimodal transport systems. However, in practice, the RTIs attrition of misplacement, missing, or other means frequently happens and has been a major cause of financial losses. Ilic et al. estimated that the annual attrition rate of returnable trays is around 10% [1], and Breen showed that about 140 million pounds were lost due to the loss of RTIs in the United Kingdom every year [2]. Moreover, ineffective management makes the number and time at which RTIs are returned uncertain and unpredictable, which forces workers to wait and suspend their ongoing tasks. This not only increases the variance of workers' workload but also causes financial loss from the suspension of tasks. In addition, the management of RTIs is a resource-intensive activity that requires workers to perform a huge amount of simple and repetitive (thus low-value-added) tasks like categorizing and enumerating

RTIs, which can severely increase labor costs. Therefore, it is urgently needed to improve the management effectiveness and efficiency of RTIs in multimodal transport systems.

Internet of Things (IoT) provides an opportunity to solve these problems, which is proved to be effective in realizing real-time tracking and tracing of various objects in diverse fields [3–8]. However, in the practices of logistics and manufacturing industries, especially in small and medium enterprises (SMEs), IoT technologies have not been widely adopted. Besides the technical burdens for varied scenarios, the high cost of IoT applications should be another obstruction [9]. Meanwhile, considering the high investment and maintenance cost of IoT systems, it is common that not all the logistics scenarios are suitable for adopting IoT systems or could benefit from them. Therefore, many scholars have made detailed cost–benefit analyses on adopting IoT technologies in diverse fields, such as retail stores [10], supply chain management [11], and construction systems [12].

An important task in managing RTI systems is the forecasting of RTI returns. The first paper that proposed a method for forecasting RTI returns is Goh and Varaprasad [13], who developed a statistical methodology for analyzing the life-cycle of returnable containers assuming they are subject to damage and loss. Kelle and Silver developed four different procedures to forecast the expected demands and expected returns of RTIs, based on varying amounts of information [14]. Bojkow developed a simulation model for estimating the average number of trips made by an RTI during its life cycle [15]. Other related work in this area included Kelle and Silver [16], Buchanan and Abad [17], and Chew et al. [18].

The above research focused on forecasting RTIs returns, thereby assisting decision makers in planning the purchase, distribution, and return of RTIs. Optimizing the operation process of the RTIs supply chain by integrating resources to achieve cost reduction and efficiency increase in RTIs management is also one of the important research directions. Del Castillo and Cochran developed one of the first models for managing RTI systems in a closed-loop supply chain [19]. They focused on the interaction between the production of finished products and the handling and distribution of RTIs. Kim et al. developed an analytical model of a two-stage supply chain, where RTIs were used to ship deteriorating products from supplier to buyer. They assumed stochastic return time of RTIs, and finished products that deteriorate during delivery delays [20]. The results indicated that the model developed by researchers could improve coordination in the system. Glock and Kim developed a mathematical model of a single-vendor–single-buyer supply chain, where RTIs were used to transport finished products [21]. The results indicated that the model can help practitioners determine a delivery policy and optimal lot sizes for both finished products and RTIs that minimize the total cost. Glock and Kim studied alternative safety measures that can decrease the likelihood of stockouts, namely: I) RTI safety return times, II) RTI safety stocks, and III) a combination of both measures [22]. The results described the conditions the safety measures are suitable for. Cobb proposed an inventory control model for used, inspected, repaired, and purchased RTIs in a closed-loop supply chain. The results showed that the minimum cost solution was obtained when inspection and repair runs began simultaneously [23]. Achamrah et al. proposed an artificial immune system-based algorithm enhanced with deep reinforcement learning for optimizing RTI flows in a two-level closed-loop supply chain [24]. Tornese et al. gave a detailed review of the RTIs management in the supply chain [25]. Liu et al. developed a decision support framework to optimize the distribution flows and dispatching vehicle routes by the use of a two-stage solution process [26]. Fan et al. developed an inventory model to minimize the total cost of the RTIs management system. They further analyzed the optimal decision of the closed-loop supply chain of RTI and compared the case where the retailer invests in reducing RTI loss and the case where the retailer does not [27]. Zhang et al. developed an improved bi-objective mixed-integer liner program to optimize the total profit of integrated multi-period closed-loop food supply-chain planning problem with returnable transport items [28]. Other related work in this area included Tsiliyannis [29], Atamer et al. [30], Goudenege et al. [31], Bottani et al. [32], Santos et al. [33], Mensendiek [34], Ni et al. [35],

Hariga et al. [36], Zhang et al. [37], Na et al. [38] and so on. These studies were aimed at providing coordinating management methods of RTI systems for practitioners.

Although extensive knowledge has been accumulated through previous research, most of them focused on how to accurately predict the return rate of RTIs and how to carry out collaborative optimization to improve the management efficiency of RTIs, which cannot be directly used for the intelligent transformation of RTIs management system, especially for multimodal transport systems. Rare research has been carried out on RTIs management in multimodal transport systems, and challenges for cost–benefit analysis still exist. IoT-enabled RTIs management system in multimodal transport system has not been widely accepted due to a lack of effective decision models to assist enterprises in the intelligent transformation of RTIs management system. The challenges are as follows: (1) RTIs management is coupled with the logistics operational process that makes its cost analysis difficult. (2) In multimodal transport, the operation and management process of RTIs are numerous and complex, and there are many factors affecting the management cost of RTIs. Therefore, how to build decision models under different schemes has become a challenge.

Taking the above challenges into consideration, this work proposed decision models to help enterprises determine the conditions under which adopting IoT-RTIs management system is economical. Three questions should be answered for developing a decision model: (1) What is the operation process of RTIs in multimodal transport system? (2) How to develop decision models according to different operation process? (3) What are the main factors affecting the benefits? Focusing on these questions through extensive field studies in collaborative logistics service providers in multimodal transport systems, this research first proposed three typical schemes of RTIs management as the targeted scenarios. Then, the cost–benefit analyses of them are conducted while the decision models on whether to adopt IoT technologies are built. Finally, based on the decision models, the main factors affecting the application of IoT-RTIs management system are studied by numerical analysis, based on which several managerial implications are presented.

The main contributions of this research are as follows:

(1) Through extensive field studies in collaborative logistics service providers in multi­modal transport systems, this research summarized the operation process of RTIs into three typical schemes.

(2) Based on three typical schemes, this work proposed decision models to help enter­prises determine the conditions under which adopting IoT-RTIs management system is economical.

(3) Based on decision models, this research studied the main factors affecting the applica­tion of IoT-RTIs management systems, based on which several managerial implica­tions are presented.

The research idea of this paper is shown in Figure 1. The rest of the paper is organized as follows. Section 2 describes the problem of RTIs management and some necessary assumptions for cost–benefit analysis. Meanwhile, the cost–benefit models for the three management schemes are developed in Section 2. The numerical analysis is given in Sec­tion 3. Finally, the discussion and conclusions are presented in Sections 4 and 5, respectively.

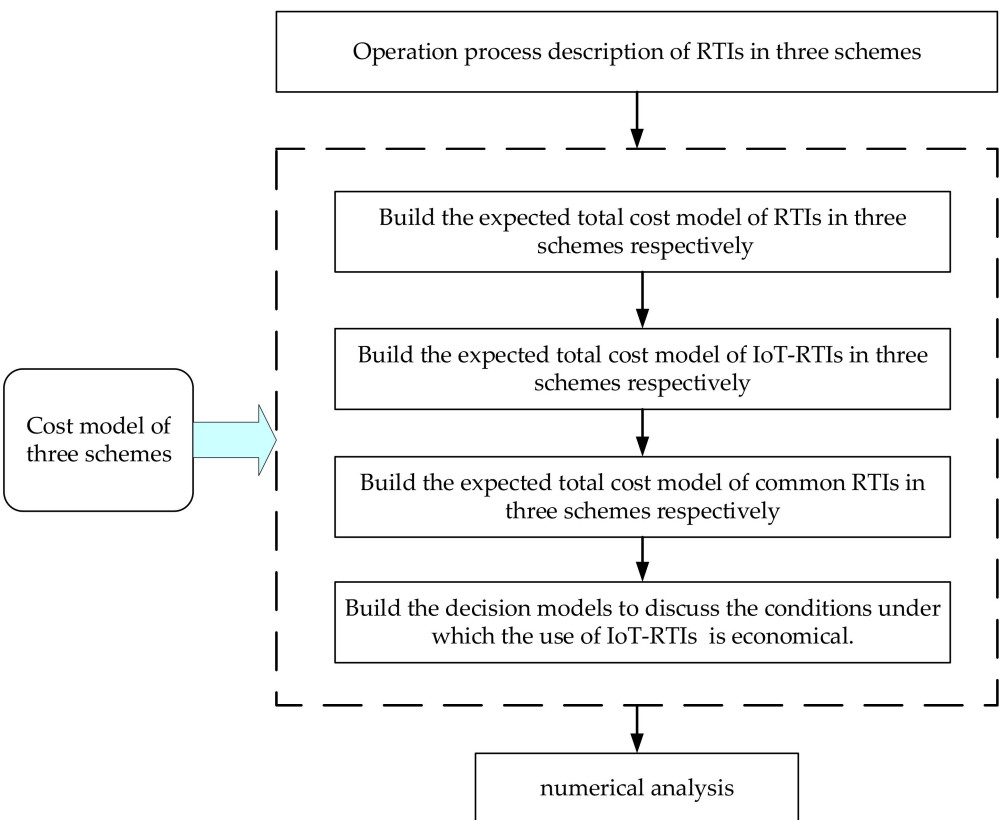

**Figure 1.** Research idea of this paper.

## 2. Materials and Methods

In this section, we first describe the notations used in modeling, then present three possible RTIs management models in multimodal transport systems and describe the workflow of each model in detail, and finally make some necessary assumptions before building the total cost models.

The following nomenclature will be used throughout the paper. Most of the notation has been retained from Kim and Glock [39].

### 2.1. Model Description

In multimodal transport systems, the circulation of RTIs is usually a closed circle in which they are sent from the inventory of RTIs to the warehouse center (WC), used to repack products at the warehouse center, sent to a destination with products, and returned to the inventory of RTIs after the products are extracted at the destination warehouse (DW). Due to the lack of visibility in the entire logistics cycle, the management of RTIs is very chaotic, and problems such as misplacement, damage, and loss often occur. Therefore, when enterprises purchase RTIs, there are three possible procurement schemes: In Section 2.3.1, RTIs must be procured for each cycle to maintain their supply. In Section 2.3.2, the lost RTIs are replenished in one batch after multiple cycles rather than once each cycle. In Section 2.3.3, RTIs are also procured after multiple cycles, except the RTIs sent from the IC to the WC are allowed to be fewer than d. In the following, we describe the workflow of each scheme in detail.

In Section 2.3.1, RTIs must be procured for each cycle to maintain their supply. The freight forwarder sends the products to the warehouse of the logistics service provider (LSP). When DW needs products, it makes a call for products to the WC, which then sends a call for RTIs to the inventory center of the RTIs (IC), to which the IC will respond by sending the correct number of RTIs. The material flow of the RTIs is described in Figure 2. Supposing d RTIs are needed in unit time, the IC then sends out d RTIs. Only $\gamma d$ RTIs

can be directly sent to the WC, while $(1 - \gamma)d$ RTIs are delivered to the inventory center of repairable RTIs (IRC) for inspection maintenance. When fully repaired, the latter are kept ready in the IRC until called by the WC. After receiving the RTIs, the WC uses them to repack and send products to the destination, which extracts the products and returns the RTIs to the IC. Attrition of RTIs generally takes place during the trip from the DW to the IC. Suppose $(1 - \alpha)d$ RTIs are lost, then $\alpha d$ RTIs are sent out from the DW. Before being moved into storage, these $\alpha d$ RTIs are first inspected at the IC, from which the unrepairable $(1 - \beta)\alpha d$ RTIs are discarded, and the remaining $\alpha\beta d$ RTIs are delivered to the IC warehouses to be ready for the call from the next cycle. Meanwhile, the IC procures $(1 - \alpha\beta)d$ RTIs, and a new cycle begins.

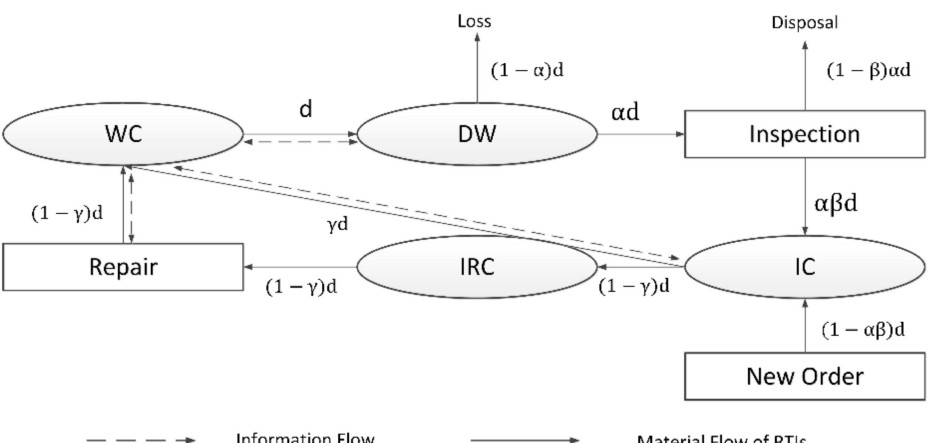

**Figure 2.** The material flow of RTIs in Section 2.3.1.

Section 2.3.2 follows the same procedure as Section 2.3.1 in each cycle, except lost RTIs are replenished in one batch after multiple cycles rather than once each cycle. The material flow of RTIs is described in Figure 3. Similar to Section 2.3.1, $(1 - \alpha\beta)d$ RTIs are lost in each cycle, and further supposes the IC makes a procurement round every $n_2$ cycles. Thus to ensure the IC can supply the WC with d RTIs each cycle, $n_2(1 - \alpha\beta)d$ RTIs must be replenished by procurement. That is to say, there are $n_2(1 - \alpha\beta)d + d$ total RTIs at the start in the IC, and every cycle reduces them by $(1 - \alpha\beta)d$ RTIs. After $n_2$ cycles, the count is reduced to d, while the IC procures $n_2(1 - \alpha\beta)d$ RTIs, which returns the total number to $n_2(1 - \alpha\beta)d + d$.

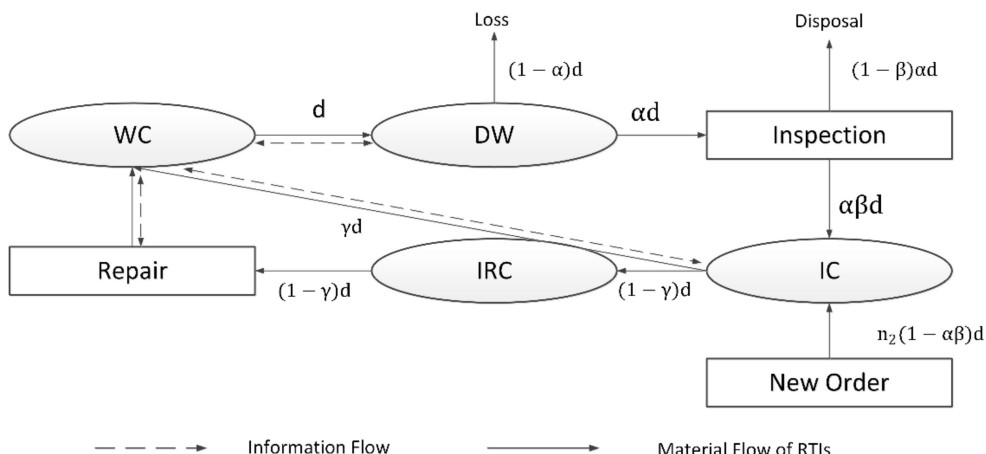

**Figure 3.** The material flow of RTIs in Section 2.3.2.

In Section 2.3.3, RTIs are also procured after multiple cycles, except the RTIs sent from the IC to the WC are allowed to be fewer than d. Since fewer products than optimal can

be sent each cycle, a financial loss is incurred, denoted as penalty cost (PC). The material flow of RTIs is described in Figure 4. Suppose a procurement is made every $n_3$ cycles, the following should happen in the $n_3$-th cycle: it starts with $\alpha^{n_3-1}\beta^{n_3-1}d$ RTIs in the IC, of which $\alpha^{n_3-1}\beta^{n_3-1}\gamma d$ RTIs are directly sent to the WC, and $\alpha^{n_3-1}\beta^{n_3-1}(1-\gamma)d$ RTIs are sent to the IRC for maintenance and waiting to be called by the WC. The WC repacks and sends the products to the DW, which extracts products and sends RTIs back to the IC. During this trip, $\alpha^{n_3-1}\beta^{n_3-1}(1-\alpha)d$ RTIs are lost, then $\alpha^{n_3}\beta^{n_3-1}d$ RTIs are sent out from the DW. Before moved into storage, these $\alpha^{n_3}\beta^{n_3-1}d$ RTIs are first inspected at the IC, from which $\alpha^{n_3}\beta^{n_3-1}(1-\beta)d$ RTIs are unrepairable and discarded, leaving $\alpha^{n_3}\beta^{n_3}d$ RTIs for the the IC warehouse. After this cycle, $(1-\alpha^{n_3}\beta^{n_3})d$ RTIs are purchased to replenish the RTI count to d, and the process starts anew.

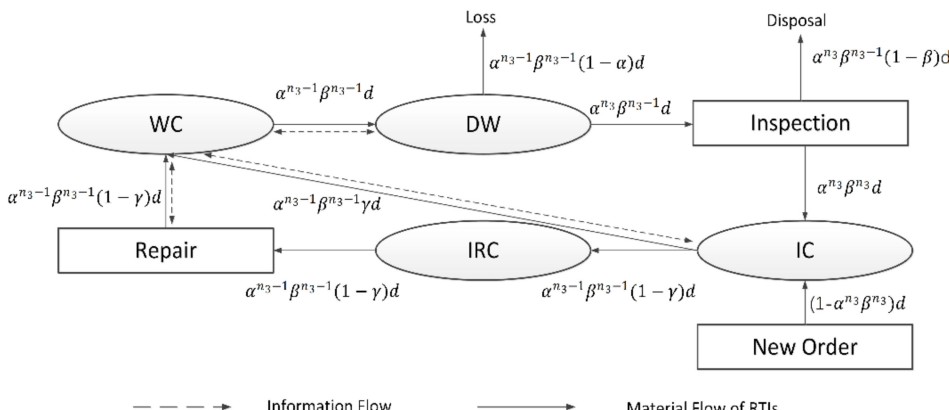

**Figure 4.** The material flow of RTIs in Section 2.3.3.

In Section 2.3.1, the IC can satisfy the need of the WC by purchasing new RTIs at the end of each cycle, and each purchase is small enough to not tie up too much cash from the IC. However, this scheme is very demanding on the ability to communicate and coordinate between the LSP and the RTIs manufacturer. Section 2.3.2 only requires one purchase after several cycles, which is less demanding, but each purchase must tie up considerable cash. Both schemes prioritize maintaining the RTIs count, making them suitable for firms that have strict requirements on the number of RTIs they need and can suffer severe losses when the requirements are not met. Section 2.3.3 also makes one purchase after more than one cycle, but the purchase only replenishes enough RTIs demanded by the first cycle, leaving fewer RTIs for all cycles after the first until the resupply. This scheme is less demanding and does not require too much cash for each purchase, and can be viable if the losses from not having enough RTIs are acceptable.

It also can be seen from the above workflow that the lack of effective visible monitoring methods will directly affect the return rate and damage rate of the RTIs, and further affect the management cost of the RTIs. Adopting IoT technology in the RTIs management system can improve the visibility of RTIs management, increase the return rate of RTIs, and reduce their damage rate, thereby improving the management efficiency of RTIs. However, the IoT-RTIs management system needs an intelligent upgrade of related equipment, which increases the cost of transformation. Therefore, in Section 2.3, we will propose decision models to discuss in detail the conditions under which the use of IoT-RTIs management systems is economical.

### 2.2. Assumptions

Apart from what has been stated above, we make the following assumptions:

1　　All RTIs must be in perfect working condition before they can be used by the WC, which is why we assume some RTIs coming out from the IC still needs to undergo maintenance. After each operation cycle, some RTIs must be repaired to ensure their

working conditions. Therefore, the return rate of RTIs should be considered when modeling.

2　　Even with IoT-RTIs management systems, the locations of RTIs can only be tracked at key nodes like the IC, IRC, WC, and DW, not throughout the entire production cycle. We assume IoT-RTIs management systems adoption can only increase the ratio of RTIs returned from the DW to the IC, not eliminate RTIs attrition. Therefore, even if IoT-enabled RTIs are used, the return rate of RTIs still needs to be considered when modeling, and the return rate should be larger than common RTIs and less than 1.

### 2.3. Cost–Benefit Analysis

In the following, we will establish management cost and decision models for three schemes. The modeling process refers to the modeling idea of EOQ and the modeling process of Kim and Glock [39].

### 2.3.1. Scheme 1

The total cost model consists of fixed and variable costs of managing RTIs as well as inventory holding costs. In this section, the implementation cost of the IoT-RTIs management system is not considered. The implementation cost is only considered when calculating the expected total cost of the IoT-RTIs management system. Thus:

$$TC_1 = FC_1 + VC_1 + HC_1 \tag{1}$$

The fixed cost of managing RTIs, $FC_1$, denotes fixed costs from RTI inspection, repair, and procurement. Such costs are related not to the number of RTIs, and only to the number of times these activities are performed. In Section 2.3.1, each cycle only involves one inspection, repair, and procurement activity each. Thus:

$$FC_1 = \frac{C_I + C_R + C_P}{T} \tag{2}$$

The variable cost of managing RTIs, $VC_1$, denotes variable costs from RTI inspection, repair, and procurement. Such costs are related to the number of RTIs. Thus:

$$VC_1 = \alpha dc_i + (1 - \gamma)dc_r + (1 - \alpha\beta)dc_p \tag{3}$$

The inventory holding costs, $HC_1$, denotes the costs from the storage process of RTIs at each key node. The inventory pattern of Section 2.3.1 is shown in Figure 5. $HC_1$ is given as follows:

$$HC_1 = \frac{1}{2}\alpha\beta dTh_u + (1 - \gamma)\left(\gamma - \frac{1}{n}\right)dTh_r + \left(\gamma^2 - \left(1 + \frac{1}{n}\right)\gamma + \frac{1}{2} + \frac{1}{2n}\right)dTh_w \tag{4}$$

$$+ \frac{1}{2n}dTh_p$$

In Equation (4), the four components respectively describe the inventory holding cost at the IC, IRC, WC, and DW. The costs are determined by first calculating the shadowed area in Figure 5, then having the corresponding number of RTIs multiplied by the storage cost of a single RTI, and finally divided by the cycle time. The total cost function can now be formulated as follows:

$$TC_1 = \frac{C_I + C_R + C_P}{T} + \alpha dc_i + (1 - \gamma)dc_r + (1 - \alpha\beta)dc_p + \frac{1}{2}\alpha\beta dTh_u$$

$$+ (1 - \gamma)\left(\gamma - \frac{1}{n}\right)dTh_r + \left(\gamma^2 - \left(1 + \frac{1}{n}\right)\gamma + \frac{1}{2} + \frac{1}{2n}\right)dTh_w + \frac{1}{2n}dTh_p \tag{5}$$

We assume the IoT-RTIs management system adoption only affects the return rate from the DW to the IC, thus in Equation (5), $\alpha$ is a random variable, and all other values

are fixed. We also use $E(\alpha)$ to denote the expectation of $\alpha$, and assume $0 \leq \alpha \leq 1$. Thus, the expected total cost of Section 2.3.1 is given as follows:

$$ETC_1 = \int_0^1 TC_1 f(\alpha) d\alpha = EFC_1 + EVC_1 + EHC_1 \tag{6}$$

where $EHC_1 = \frac{1}{2}E(\alpha)\beta dTh_u + M_1\left((1-\gamma)\left(\gamma-\frac{1}{n}\right)dTh_r + \left(\gamma^2 - \left(1+\frac{1}{n}\right)\gamma + \frac{1}{2} + \frac{1}{2n}\right)dTh_w + \frac{1}{2n}dTh_p\right)$, $EVC_1 = E(\alpha)dc_i + M_1(1-\gamma)dc_r + (M_1 - E(\alpha)\beta)dc_p$, $EFC_1 = \frac{C_I + C_R + C_P}{T}M_1$, $M_1 = \int_0^1 f(\alpha)d\alpha$ and $E(\alpha) = \int_0^1 \alpha f(\alpha)d\alpha$.

In multimodal transport systems, the key to the construction of the IoT-RTIs management system is the intelligent transformation of RTIs. Therefore, compared with RTIs, the procurement cost of IoT-RTIs will increase. But on the other hand, the use of IoT-RTIs management systems can monitor the working status of RTIs and track their location, which can increase the return rate of RTIs, reduce the purchase quantity of RTIs, and thereby reduce the management cost of RTIs. Therefore, if the total cost with IoT-RTIs management system adoption is lower than that without it, the adoption would be profitable and worth conducting, and vice versa. Here $c_{p,IoT}$ denotes the purchase price of one IoT-RTI, and $c_{p,N}$ denotes the purchase price of an RTI. Similarly, hereafter the subscript *IoT* will denote the IoT-RTIs management system, and subscript $N$ will denote a common RTIs management system. Thus, the expected total cost of the IoT-RTIs management system is given as follows:

$$ETC_{1,IoT} = EFC_{1,IoT} + E(\alpha_{IoT})dc_i + M_{1,IoT}(1-\gamma)dc_r$$
$$+ (M_{1,IoT} - E(\alpha_{IoT})\beta)dc_{p,IoT} + EHC_{1,IoT} + ETRC_{IoT} \tag{7}$$

where $ETRC_{IoT} = C_{TR}M_1$ represents the expected implementation cost of the IoT-RTIs management system in each cycle.

The expected total cost of common RTIs management systems is given as follows:

$$ETC_{1,N} = EFC_{1,N} + E(\alpha_N)dc_i + M_{1,N}(1-\gamma)dc_r$$
$$+ (M_{1,N} - E(\alpha_N)\beta)dc_{p,N} + EHC_{1,N} \tag{8}$$

When the expected total cost of IoT-RTIs management system is less than or equal to the expected total cost of common RTIs management systems, the use of IoT-RTIs management systems is economical. Thus:

$$ETC_{1,IoT} \leq ETC_{1,N} \tag{9}$$

From Equation (9), we can derive the following conditions for $c_{p,IoT}$:

$$c_{p,\text{IoT}} \leq \bar{c}_{1,IoT} = \frac{ETC_{1,N} - (EFC_{1,\text{IoT}} + E(\alpha_{\text{IoT}})dc_i + M_{1,\text{IoT}}(1-\gamma)dc_r + EHC_{1,\text{IoT}} + ETRC_{IoT})}{(M_{1,\text{IoT}} - E(\alpha_{\text{IoT}})\beta)d} \tag{10}$$

From Equation (10), we conclude that if the unit price of IoT-RTI is less than or equal to $\bar{c}_{1,IoT}$, it is beneficial for the enterprise to use IoT-RTI in the system. If not, the use of IoT-RTI is not economical and common RTIs should be used in the system.

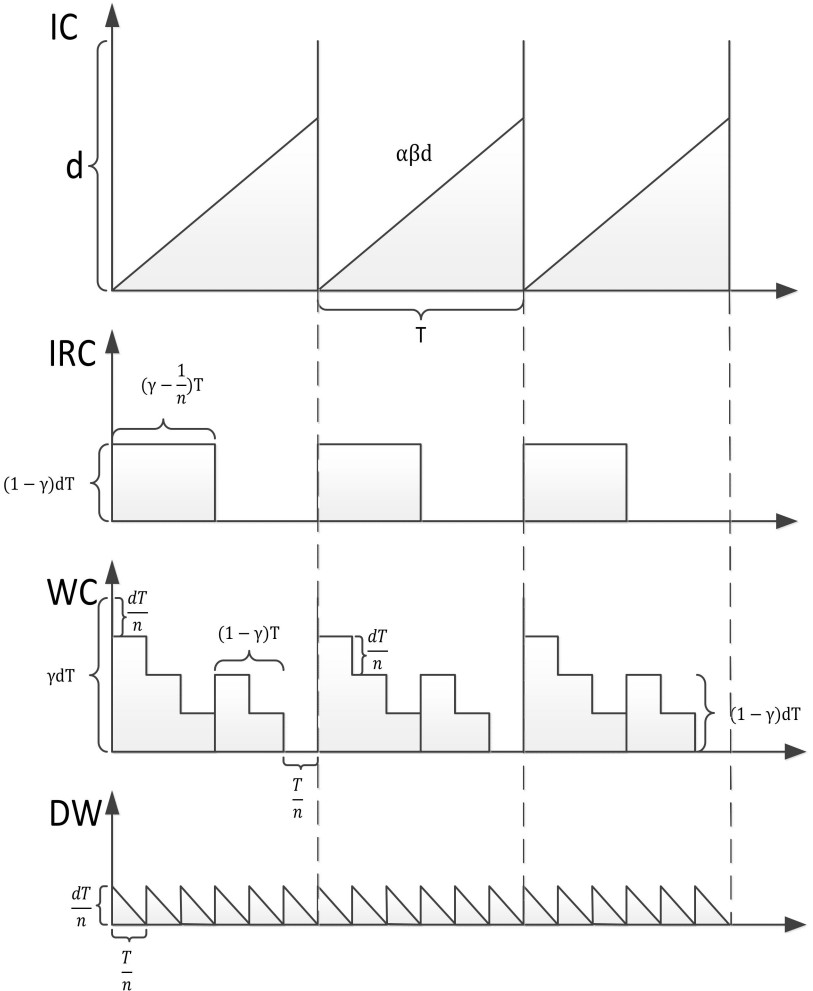

**Figure 5.** Inventory patterns of RTIs at each stage for Section 2.3.1.

### 2.3.2. Scheme 2

In Section 2.3.2, a procurement is made every $n_2$ cycles, and any "total cost" is the sum of costs over $n_2$ cycles. Similar to Section 2.3.1, the total cost model consists of fixed and variable costs of managing RTIs as well as inventory holding costs. Thus:

$$TC_2 = FC_2 + VC_2 + HC_2 \tag{11}$$

The fixed cost of managing RTIs, $FC_2$, denotes fixed costs from RTIs inspection, repair, and procurement over $n_2$ cycles. We assume inspection and repair are performed each cycle, and one purchase is made every $n_2$ cycles. Thus:

$$FC_2 = \frac{(C_I + C_R)n_2 + C_P}{T} \tag{12}$$

The variable cost of managing RTIs, $VC_2$, denotes variable costs from RTIs inspection, repair, and procurement over $n_2$ cycles. Such costs are related not only to the number of RTIs, but also to the number of times the activities are performed. We assume there are d RTIs in the IC after the $n_2$-th cycle, and as shown in Figure 3, $(1 - \alpha\beta)$d RTIs are lost in each cycle, which means $n_2(1 - \alpha\beta)d$ RTIs need to be purchased. Thus:

$$VC_2 = (\alpha d c_i + (1 - \gamma)d c_r)n_2 + n_2(1 - \alpha\beta)d c_p \tag{13}$$

The inventory holding costs, $HC_2$, denotes the costs from the storage process of RTIs at each key node. The inventory pattern of Section 2.3.2 is shown in Figure 6. Let $IC_{2t}$, $IRC_{2t}$, $WC_{2t}$, $DW_{2t}$ be the respective total storage costs of the IC, IRC, WC, and DW over $n_2$ cycles, and $HC_2$ be the total storage cost of all nodes over $n_2$ cycles. Thus:

$$IC_{2t} = \left( n_2 \frac{1}{2} \alpha \beta dT + (1 - \alpha \beta) dT (1 + 2 + \cdots + n_2) \right) h_u$$

$$= \frac{n_2}{2} (\alpha \beta + (n_2 + 1)(1 - \alpha \beta)) dT h_u \tag{14}$$

$$IRC_{2t} = n_2 (1 - \gamma) \left( \gamma - \frac{1}{n} \right) dT h_r \tag{15}$$

$$WC_{2t} = \frac{1}{2} n_2 \left( \gamma \left( \gamma - \frac{1}{n} \right) + (1 - \gamma) \left( 1 - \gamma + \frac{1}{n} \right) \right) dT h_w \tag{16}$$

$$DW_{2t} = \frac{1}{2n} n_2 dT h_p \tag{17}$$

$$HC_2 = IC_{2t} + IRC_{2t} + WC_{2t} + DW_{2t} \tag{18}$$

Thus, the expected total cost of Section 2.3.2 is given as follows:

$$ETC_2 = \int_0^1 TC_2 f(\alpha) d\alpha = EFC_2 + EVC_2 + EHC_2 \tag{19}$$

where $EHC_2 = \frac{n_2}{2} (E(\alpha)\beta + (n_2 + 1)(M_1 - E(\alpha)\beta)) dT h_u + M_1 (IRC_{2t} + WC_{2t} + DW_{2t})$, $EVC_2 = (E(\alpha)dc_i + M_1(1 - \gamma)dc_r)n_2 + n_2(M_1 - E(\alpha)\beta)dc_p$ and $EFC_2 = M_1 \frac{(C_I + C_R)n_2 + C_P}{T}$.

The expected total cost of the IoT-RTIs management system is given as follows:

$$ETC_{2,IoT} = EFC_{2,IoT} + (E(\alpha_{IoT})dc_i + M_{1,IoT}(1 - \gamma)dc_r)n_2$$

$$+ n_2(M_{1,IoT} - E(\alpha_{IoT})\beta)dc_{p,IoT} + EHC_{2,IoT} + ETRC_{IoT} \tag{20}$$

where $ETRC_{IoT} = C_{TR}M_1$ represents the expected implementation cost of the IoT-RTIs management system in each cycle.

The expected total cost of a common RTI management system is given as follows:

$$ETC_{2,N} = EFC_{2,N} + (E(\alpha_N)dc_i + M_{1,N}(1 - \gamma)dc_r)n_2$$

$$+ n_2(M_{1,N} - E(\alpha_N)\beta)dc_{p,N} + EHC_{2,N} \tag{21}$$

When the expected total cost of an IoT-RTIs management system is less than or equal to the expected total cost of a common RTIs system, the use of an IoT-RTIs management system is economical. Thus:

$$ETC_{2,IoT} \leq ETC_{2,N} \tag{22}$$

From this inequality, we can derive the following conditions for $c_{p,IoT}$:

$$c_{p,IoT} \leq \bar{c}_{2,IoT} = \frac{ETC_{2,N} - (EFC_{2,IoT} + EHC_{2,IoT} + ETRC_{IoT} + (E(\alpha_{IoT})dc_i + M_{1,IoT}(1 - \gamma)dc_r)n_2)}{n_2(M_{1,IoT} - E(\alpha_{IoT})\beta)d} \tag{23}$$

We conclude that if the unit price of IoT-RTI is less than or equal to $\bar{c}_{2,IoT}$, it is beneficial for the enterprise to use IoT-RTIs in the system. If not, the use of IoT-RTIs is not economical and the common RTI should be used in the system.

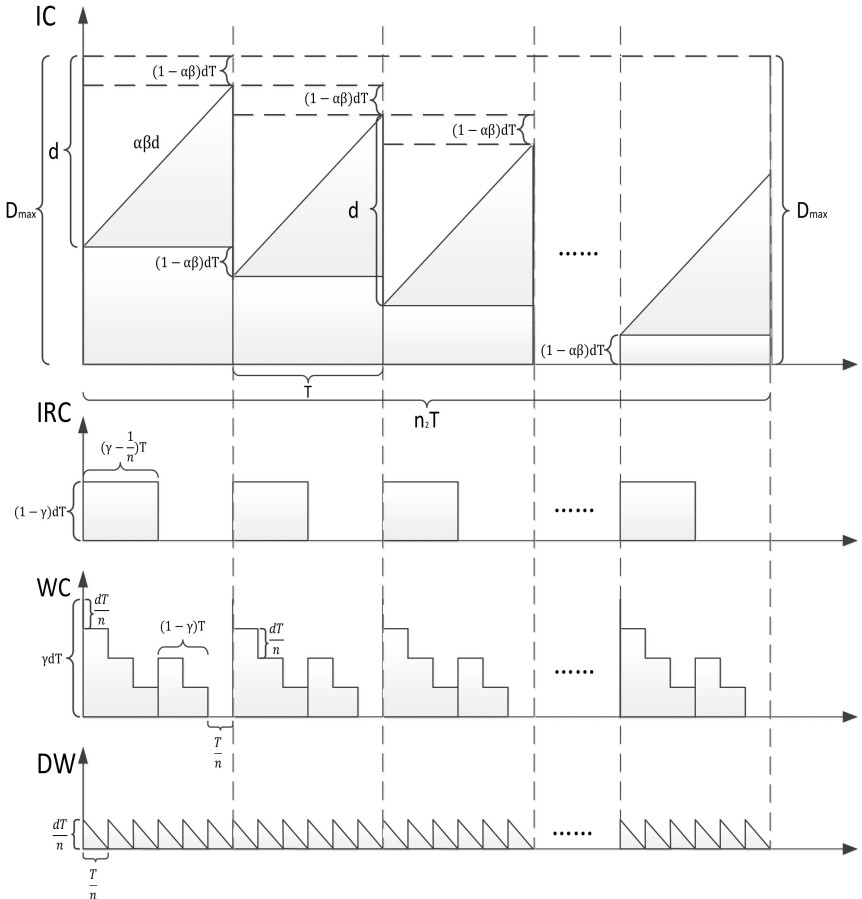

**Figure 6.** Inventory patterns of RTIs at each stage for Section 2.3.2.

### 2.3.3. Scheme 3

In Section 2.3.3, purchases are also separated by multiple cycles, and any "total cost" refers to the sum of costs over several cycles. Different from Section 2.3.2, the total cost model consists of fixed cost, variable cost, inventory holding cost, and penalty cost. Thus:

$$TC_3 = FC_3 + VC_3 + HC_3 + PC_t \tag{24}$$

The fixed cost of managing RTIs, $FC_3$, denotes fixed costs from RTIs inspection, repair, and procurement over $n_3$ cycles. We assume inspection and repair are performed once each cycle, and one purchase is made every $n_3$ cycles. Thus, $FC_3$ is calculated as follows:

$$FC_3 = \frac{(C_I + C_R)n_3 + C_P}{T} \tag{25}$$

The variable cost of managing RTIs, $VC_3$, denotes variable costs from RTIs inspection, repair, and procurement over $n_3$ cycles. Such costs are related not only to the number of RTIs, but also to the number of times the activities are performed. Under Section 2.3.3, the RTI numbers in the IC storage and under IRC maintenance will both decrease from one cycle to the next. Thus, $VC_3$ is given as follows:

$$VC_3 = \alpha dc_i \left( k^0 + k^1 + \cdots + k^{n_3-1} \right) + (1-\gamma)dc_r \left( k^0 + k^1 + \cdots + k^{n_3-1} \right)$$

$$+ (1 - k^{n_3})dc_p = \frac{k^{n_3} - 1}{k - 1}(\alpha dc_i + (1-\gamma)dc_r) + (1 - k^{n_3})dc_p \tag{26}$$

where $k = \alpha\beta$.

The inventory holding costs, $HC_3$, denotes the costs from the storage process of RTIs at each key node. The inventory pattern of Section 2.3.3 is shown in Figure 7. Let $IC_{3t}$, $IRC_{3t}$, $WC_{3t}$, $DW_{3t}$ be the respective total storage costs of the IC, IRC, WC, and DW over $n_3$ cycles, and $HC_3$ be the total storage cost of all nodes over $n_3$ cycles. Thus:

$$IC_{3t} = \frac{1}{2}\left(k + k^1 + \cdots + k^{n_3}\right)dTh_u = \frac{1}{2}\frac{k^{n_3+1} - k}{k-1}dTh_u \tag{27}$$

$$IRC_{3t} = \left(\gamma(1-\gamma)\left(k^0 + k^2 + \cdots + k^{2(n_3-1)}\right) - \frac{1-\gamma}{n}\left(k^0 + k^1 + \cdots k^{n_3-1}\right)\right)dTh_r$$

$$= \left(\frac{k^{2n_3} - 1}{k^2 - 1}\gamma(1-\gamma) - \frac{1}{n}\frac{k^{n_3} - 1}{k-1}(1-\gamma)\right)dTh_r \tag{28}$$

$$WC_{3t} = \frac{1}{2}\left(\left(2\gamma^2 - 2\gamma + 1\right)\left(k^0 + k^2 + \cdots + k^{2(n_3-1)}\right) + \frac{1}{n}(1-2\gamma)(k^0 + k^1 +$$

$$\cdots k^{n_3-1})\right)dTh_w = \frac{1}{2}\left(\frac{k^{2n_3} - 1}{k^2 - 1}\left(2\gamma^2 - 2\gamma + 1\right) + \frac{1}{n}\frac{k^{n_3} - 1}{k-1}(1-2\gamma)\right)dTh_w \tag{29}$$

$$DW_{3t} = \frac{1}{2n}\left(k^0 + k^1 + \cdots k^{n_3-1}\right)dTh_p = \frac{1}{2n}\frac{k^{n_3} - 1}{k-1}dTh_p \tag{30}$$

$$HC_3 = IC_{3t} + IRC_{3t} + WC_{3t} + DW_{3t} \tag{31}$$

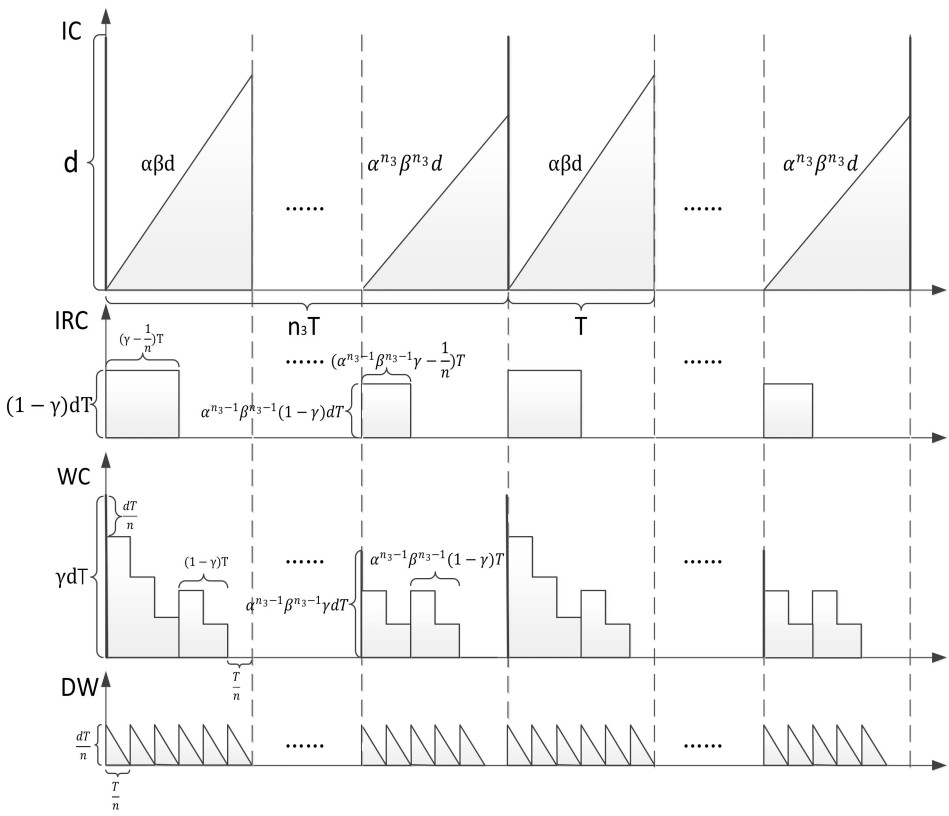

**Figure 7.** Inventory patterns of RTIs at each stage for Section 2.3.3.

Except for the first cycle after procurement, no other cycle under Section 2.3.3 can provide the WC with enough RTIs it needs. For this reason, a PC (inversely correlated to $\alpha$, $\beta$) is incurred every cycle. We can assume when $\alpha\beta = 0$, i.e., the IC has no RTIs available for WC, the PC is at its maximum $M_{max}$; and when $\alpha\beta = 1$, i.e., the IC can provide the WC with all RTIs it needs, the PC = 0. We then use a sigmoid function to approximate the relationship between the PC and $\alpha\beta$ (Figure 8).

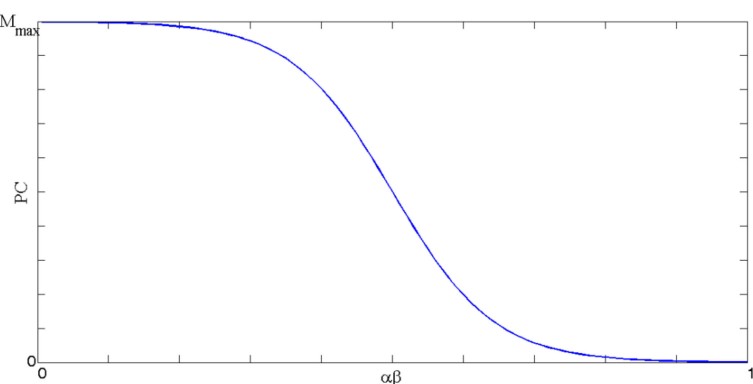

**Figure 8.** Relationship between the PC and αβ.

Let $PC_t$ be the total penalty cost of $n_3$ cycles. Thus, $PC_t$ is calculated as follows:

$$PC_t = M_{max}\left(\frac{1}{1+e^{13(k^0-0.5)}} + \frac{1}{1+e^{13(k^1-0.5)}} + \cdots + \frac{1}{1+e^{13(k^{n_3-1}-0.5)}}\right) \quad (32)$$

Therefore, the expected total cost of Section 2.3.3 is given as follows:

$$ETC_3 = \int_0^1 TC_3 f(\alpha)d\alpha = EFC_3 + EVC_3 + EHC_3 + EPC_t \quad (33)$$

where $EFC_3 = \frac{(C_I+C_R)n_3+C_P}{T}M_1, EVC_3 = M_{k2}dc_i + M_{k1}(1-\gamma)dc_r + M_{k3}dc_p +$
and $EHC_3 = \frac{1}{2}M_{k4}dTh_u + (M_{k5}\gamma(1-\gamma) - \frac{1}{n}M_{k1}(1-\gamma))dTh_r + \frac{1}{2}\left(M_{k5}(2\gamma^2 - 2\gamma + 1) + \frac{1}{n}M_{k1}(1-2\gamma)\right)dTh_w + \frac{1}{2n}M_{k1}dTh_p.$

where $EPC_t = M_{max}\int_0^1\left(\frac{1}{1+e^{13(k^0-0.5)}} + \frac{1}{1+e^{13(k^1-0.5)}} + \cdots + \frac{1}{1+e^{13(k^{n_3-1}-0.5)}}\right)f(\alpha)d\alpha.$

where $M_{k1} = \int_0^1 \frac{k^{n_3}-1}{k-1}f(\alpha)d\alpha, M_{k2} = \int_0^1 \frac{k^{n_3}-1}{k-1}\alpha f(\alpha)d\alpha, M_{k3} = \int_0^1(1-k^{n_3})f(\alpha)d\alpha,$
$M_{k4} = \int_0^1 \frac{k^{n_3+1}-k}{k-1}f(\alpha)d\alpha$ and $M_{k5} = \int_0^1 \frac{k^{2n_3}-1}{k^2-1}f(\alpha)d\alpha.$

The expected total cost of the IoT-RTIs management system is given as follows:

$$ETC_{3,IoT} = EFC_{3,IoT} + M_{k2,IoT}dc_i + M_{k1,IoT}(1-\gamma)dc_r + M_{k3,IoT}dc_{p,IoT}$$

$$+ EHC_{3,IoT} + EPC_{t,IoT} + ETRC_{IoT} \quad (34)$$

where $ETRC_{IoT} = C_{TR}M_1$ represents the expected implementation cost of IoT-RTIs management system in each cycle.

The expected total cost of a common RTIs management system is given as follows:

$$ETC_{3,N} = EFC_{3,N} + M_{k2,N}dc_i + M_{k1,N}(1-\gamma)dc_r + M_{k3,N}dc_{p,N}$$

$$+ EHC_{3,N} + EPC_{t,N} \quad (35)$$

When the expected total cost of the IoT-RTIs management system is less than or equal to the expected total cost of a common RTIs management system, the use of the IoT-RTIs management system is economical. Thus:

$$ETC_{3,IoT} \leq ETC_{3,N} \quad (36)$$

From this inequality, we can derive the following conditions for $c_{p,IoT}$:

$$c_{p,IoT} \leq \bar{c}_{3,IoT} = \frac{ETC_{3,N} - (EFC_{3,IoT} + M_{k2,IoT}dc_i + M_{k1,IoT}(1-\gamma)dc_r + EHC_{3,IoT} + EPC_{t,IoT} + ETRC_{IoT})}{M_{k3,IoT}d} \quad (37)$$

We conclude that if the unit price of IoT-RTI is less than or equal to $\bar{c}_{3,IoT}$, it is beneficial for the enterprise to use IoT-RTI in the system. If not, the use of IoT-RTI is not economical and common RTI should be used in the system.

## 3. Results

### 3.1. Parameters Setting

In this section, Matlab is used for further analysis. We assume that the return rate $\alpha$ is normally distributed with a variance of 0.003. For Equations (6), (19) and (33), ten scenarios are used, with $E(\alpha)$ respectively as 0.55, 0.6, 0.65, 0.7, 0.75, 0.8, 0.85, 0.9, 0.95, and 1. The other parameters are given as follows, and some are based on Kim and Glock [39]. The result is shown in Tables 1–4.

$$d = 1000, \ T = 0.5, \ C_I = 200, \ C_R = 2, \ C_P = 10, \ c_i = 2, \ c_r = 2, \ c_p = 10, \ h_u = 4, \ h_r = 2, \ h_w = 2, \ h_p = 2, \ C_{TR} = 1000,$$

$$h_p = 2, \ n = 10, \ \beta = 0.95, \ \gamma = 0.95, n_2 = 5, n_3 = 3, M_{max} = 10000$$

In Section 2.3.1, we assume that the return rate of the IoT-RTI management system $\alpha_{IoT}$ and common RTIs management system $\alpha_N$ ($\alpha_N \sim N(0.6, 0.003)$) are all normally distributed. With ten values for $E(\alpha_{IoT})$: 0.55, 0.6, 0.65, 0.7, 0.75, 0.8, 0.85, 0.9, 0.95, 1, and the same variance of 0.003, Equation (10) can be used to obtain $\bar{c}_{1,IoT}$. Figure 9 depicts the results of the calculation.

In Section 2.3.2, we assume that the return rate of IoT-RTIs management system $\alpha_{IoT}$ and common RTIs management system $\alpha_N$ ($\alpha_N \sim N(0.6, 0.003)$) are all normally distributed. With five possible values for $n_2$: 1,2,3,4,5, ten values for $E(\alpha_{IoT})$: 0.55, 0.6, 0.65, 0.7, 0.75, 0.8, 0.85, 0.9, 0.95, 1, and the same variance of 0.003, Equation (23) can be used to obtain $\bar{c}_{2,IoT}$. The result is shown in Figure 10.

In Section 2.3.3, we assume that the return rate of IoT-RTIs management system $\alpha_{IoT}$ and common RTIs management system $\alpha_N$ ($\alpha_N \sim N(0.6, 0.003)$) are all normally distributed. With five possible values for $M_{max}$: 2000, 4000, 6000, 8000, 10,000, ten values for $E(\alpha_{IoT})$: 0.55, 0.6, 0.65, 0.7, 0.75, 0.8, 0.85, 0.9, 0.95, 1, the same variance of 0.003, and $n_3 = 3$ (purchase new containers every three cycles), Equation (37) can be used to obtain $\bar{c}_{3,IoT}$. The result is shown in Figure 11.

### 3.2. Results

In Section 2.3.1, the calculation results are shown in Table 1 and Figure 9. As displayed in Figure 9, $ETC_1$ is inversely correlated to $E(\alpha)$, which means the total cost of the system trends towards decrease as the return rate rises. Table 1 shows that said decrease of $ETC_1$ can be attributed to the inverse correlation between $EVC_1$ and $E(\alpha)$. While $EFC_1$ is fixed, and $EHC_1$ increases with $E(\alpha)$, the increase of $EHC_1$ is too small compared to the decrease of $EVC_1$. As per Table 1, when $E(\alpha)$ rises from 0.55 to 1, $EHC_1$ grows from 1022.5 to 1450, by a delta of 427.5, compared to $EVC_1$, which lowers from 5975 to 2600, by a delta of 3375, therefore $EVC_1$ is the most important factor in the variation of $ETC_1$. We can also see from Equation (3) that when $c_i \ll c_p$, the increase of $\alpha$ leads to higher maintenance costs, but the cost of procurement decreases faster. Therefore, the main reason $E(\alpha)$ is correlated inversely to $ETC_1$ is the sharp decrease of the procurement cost.

Figure 9 shows that $E(\alpha_{IoT})$ is directly correlated to $\bar{c}_{1,IoT}$. This means a higher $E(\alpha_{IoT})$ allows a higher upper limit to the cost of a single IoT-RTI with which the IoT-RTIs management system is still profitable. When $E(\alpha_{IoT}) = 0.75$, we have $c_{p,IoT} \leq \bar{c}_{1,IoT} = 10 = c_p$. In that case, the IoT-RTIs management system is only viable when purchasing one IoT-RTI costs less than a common one. In practice, for the same type of RTI, the addition of sensors must necessarily raise its cost, therefore when $E(\alpha_{IoT}) \leq E(\alpha_N)$, using a common RTIs management system is more profitable. When $E(\alpha_{IoT}) = 0.95$, we have $\bar{c}_{1,IoT} = 23.3$. We know from Equation (10) that when $c_{p,IoT} \leq \bar{c}_{1,IoT} = 23.3$, i.e., when the cost of a single IoT-RTI is at 23.3 or below, IoT-RTIs management system adoption will be viable.

**Table 1.** With different $E(\alpha)$.

| $E(\alpha)$. | $EFC_1$ | $EVC_1$ | $EHC_1$ | $ETC_1$ |
|---|---|---|---|---|
| 0.55 | 424 | 5975 | 1022.5 | 7421.5 |
| 0.60 | 424 | 5600 | 1070 | 7094 |
| 0.65 | 424 | 5225 | 1117.5 | 6766.5 |
| 0.70 | 424 | 4850 | 1165 | 6439 |
| 0.75 | 424 | 4475 | 1212.5 | 6111.5 |
| 0.80 | 424 | 4100 | 1260 | 5784 |
| 0.85 | 424 | 3725 | 1307.5 | 5456.5 |
| 0.90 | 424 | 3350 | 1355 | 5129 |
| 0.95 | 424 | 2975 | 1402 | 4801 |
| 1.00 | 424 | 2600 | 1450 | 4474 |

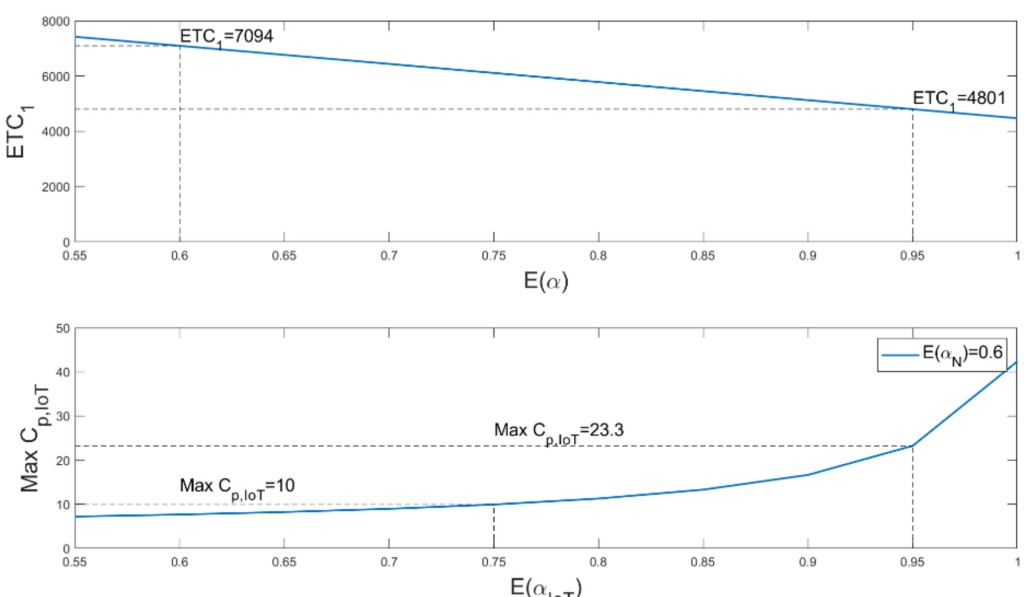

**Figure 9.** Trends of $ETC_1$ with respect to $E(\alpha)$ and trends of $\bar{c}_{1,IoT}$ with respect to $E(\alpha_{IoT})$.

In Section 2.3.2, the calculation results are shown in Table 2 and Figure 10. From Table 2, it can be seen that $E(\alpha)$ is inversely correlated to $ETC_2$. On the one hand, this is explained by the higher $E(\alpha)$ reducing the number of RTIs lost over $n_2$ cycles, which lowers the procurement cost. Equation (13) shows that while higher $E(\alpha)$ increases the maintenance cost, it is far cheaper to maintain an RTI than buying a new one, hence, the increase of $E(\alpha)$ decreases $EVC_2$. On the other hand, the increase of $E(\alpha)$ reduces the total number of RTIs required over $n_2$ cycles, which reduces the number of RTIs to be kept in the IC storage, and lowers its storage cost. Equations (15)–(17) show that $E(\alpha)$ does not affect the storage costs at the IRC, WC, and DW, hence the total storage cost $EHC_2$ is inversely correlated to $E(\alpha)$. Table 2 and Equation (12) also show that $EFC_2$ is not correlated to $E(\alpha)$. Therefore, the main factors contributing to $ETC_2$ are the procurement of new RTIs and inventory holding costs at the IC.

From Figure 10, we know that when the value of $n_2$ is given, $\bar{c}_{2,IoT}$ is directly correlated to $E(\alpha_{IoT})$. This means a higher $E(\alpha_{IoT})$ allows a higher upper limit to the cost of a single IoT-RTI with which IoT-RTIs management system adoption is still profitable. Meanwhile, when $E(\alpha_N)$ and $E(\alpha_{IoT})$ are given, $\bar{c}_{2,IoT}$ is directly correlated to $n_2$. As shown in Figure 10, when $E(\alpha_N) = 0.6$ and $E(\alpha_{IoT}) = 0.95$, if $n_2 = 1$, $\bar{c}_{2,IoT} = 30.1$; if $n_2 = 5$, $\bar{c}_{2,IoT} = 43.7$. This means, with $E(\alpha_N)$ and $E(\alpha_{IoT})$ given, a higher $n_2$ allows a higher upper limit to the cost of a single IoT-RTI with which IoT-RTIs management system adoption is still profitable.

**Table 2.** With different $E(\alpha)$.

| $E(\alpha)$. | $EFC_2$ | $EVC_2$ | $EHC_2$ | $ETC_2$ |
|---|---|---|---|---|
| 0.55 | 2040 | 29,875 | 19,438 | 51,353 |
| 0.60 | 2040 | 28,000 | 18,250 | 48,290 |
| 0.65 | 2040 | 26,125 | 17,063 | 45,228 |
| 0.70 | 2040 | 24,250 | 15,875 | 42,165 |
| 0.75 | 2040 | 22,375 | 14,688 | 39,103 |
| 0.80 | 2040 | 20,500 | 13,500 | 36,040 |
| 0.85 | 2040 | 18,625 | 12,313 | 32,978 |
| 0.90 | 2040 | 16,750 | 11,125 | 29,915 |
| 0.95 | 2040 | 14,875 | 9938 | 26,853 |
| 1.00 | 2040 | 13,000 | 8750 | 23,790 |

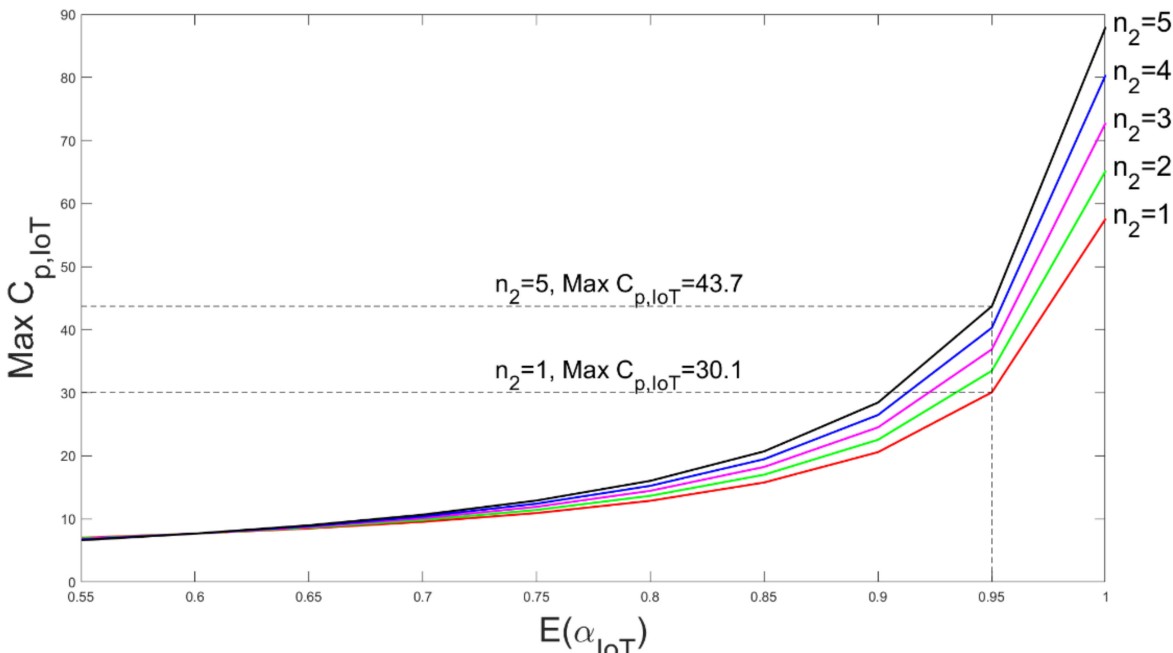

**Figure 10.** Trends of $\bar{c}_{2,IoT}$ with respect to $E(\alpha_{IoT})$ and $n_2$.

In Section 2.3.3, the calculation results are shown in Table 3 and Figure 11. From Table 3, we see $E(\alpha)$ is inversely correlated to $ETC_3$. On the one hand, this is because a higher $E(\alpha)$ reduces the number of RTIs lost over $n_3$ cycles, which lowers the procurement cost. Equation (26) shows that while higher $E(\alpha)$ increases the maintenance costs, it is far cheaper to maintain an RTI than buying a new one, hence, the increase of $E(\alpha)$ decreases $EVC_3$. On the other hand, Table 3 shows that as the increase of $E(\alpha)$ leads to less RTIs lost in each cycle, the $EPC_t$ is significantly lowered as a result. Even as $EHC_3$ increases with $E(\alpha)$ as seen in Table 3 and Equation (31), this increase is remarkably smaller than the decrease of $EVC_3$ and $EPC_t$, and $EFC_3$ is not affected by $E(\alpha)$, as seen in Table 3 and Equation (25). Therefore, the main contributing factors to $ETC_3$ are the procurement cost and penalty cost.

As is shown in Figure 11, when $n_3$ is given, $\bar{c}_{3,IoT}$ is directly correlated to $E(\alpha_{IoT})$, which means a higher $E(\alpha_{IoT})$ allows a higher upper limit to the cost of a single IoT-RTI with which IoT-RTIs management system adoption is still profitable. Meanwhile, $\bar{c}_{3,IoT}$ increases with increasing $M_{max}$. This is because, with a low $M_{max}$ the penalty cost from not having enough RTIs does not strongly affect the total costs, but higher $M_{max}$ leads to greater loss from the lack of RTIs. That is to say, with $E(\alpha_N)$ and $E(\alpha_{IoT})$ given, a higher $M_{max}$ makes IoT-RTIs management system adoption better. When $M_{max} = 2000$, $E(\alpha_{IoT}) = 0.95$, we have $\bar{c}_{3,IoT} = 17.4$, when $M_{max} = 10,000$, we have $\bar{c}_{3,IoT} = 51.4$.

**Table 3.** With different E($\alpha$).

| E($\alpha$). | $EFC_3$ | $EVC_3$ | $EHC_3$ | $EPC_t$ | $ETC_3$ |
|---|---|---|---|---|---|
| 0.55 | 1232 | 10,549 | 1612 | 13,570 | 26,963 |
| 0.60 | 1232 | 10,422 | 1795 | 11,906 | 25,355 |
| 0.65 | 1232 | 10,244 | 1998 | 10,088 | 23,562 |
| 0.70 | 1232 | 10,009 | 2220 | 8083 | 21,544 |
| 0.75 | 1232 | 9713 | 2465 | 5943 | 19,353 |
| 0.80 | 1232 | 9351 | 2732 | 3911 | 17,226 |
| 0.85 | 1232 | 8916 | 3025 | 2303 | 15,476 |
| 0.90 | 1232 | 8405 | 3344 | 1250 | 14,231 |
| 0.95 | 1232 | 7812 | 3691 | 656 | 13,391 |
| 1.00 | 1232 | 3583 | 2025 | 181 | 7021 |

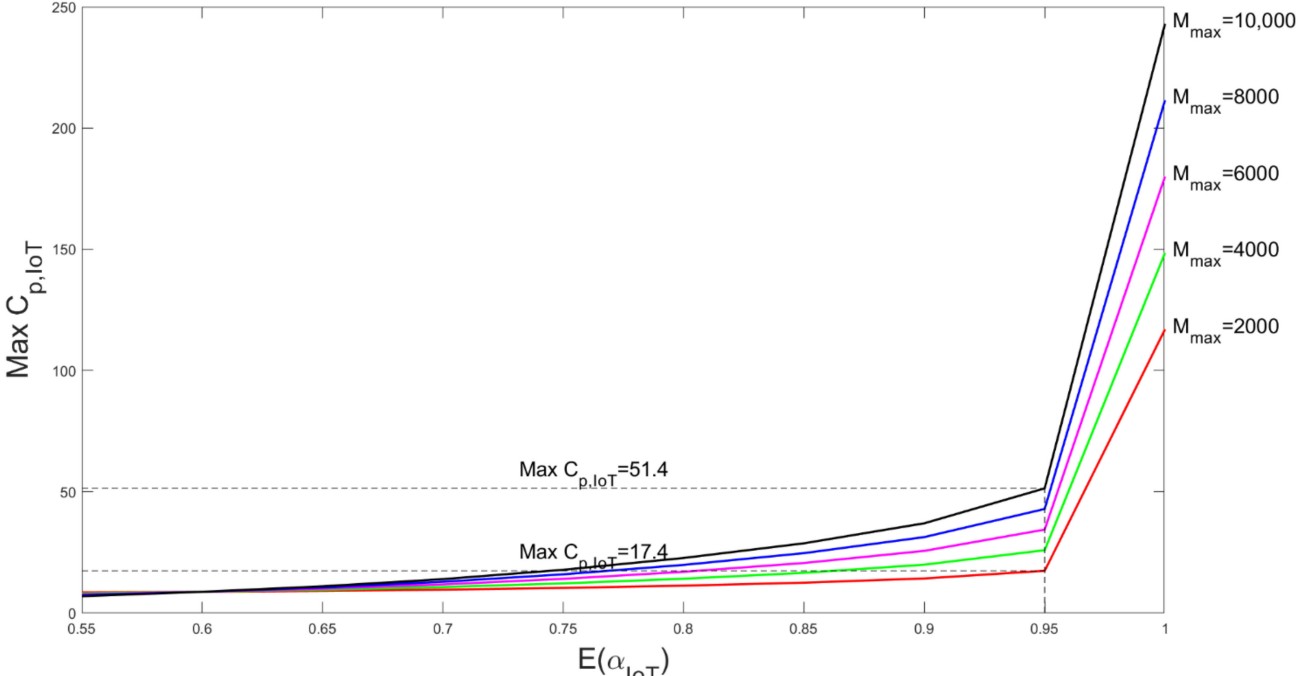

**Figure 11.** Trends of $\bar{c}_{3,\text{IoT}}$ with respect to E($\alpha_{IoT}$) and $M_{max}$.

Take Section 2.3.1 as an example, this paper will further analyze the trends of $c_{p,IoT}$ with respect to $c_p$ and $C_{TR}$. We assume that the return rate of IoT-RTIs $\alpha_{\text{IoT}}(\alpha_{IoT} \sim N(0.95, 0.003))$ and common RTIs $\alpha_N$ ($\alpha_N \sim N(0.6, 0.003)$) are all normally distributed. With ten values for $C_{TR}$: 200, 400, 600, 800, 1000, 1200, 1400, 1600, 1800, 2000, and five values for $c_p$: 10, 20, 30, 40, and 50. The analysis results are shown in Table 4 and Figure 12.

From Table 4 and Figure 12, we know that when the value of $c_p$ is given, $c_{p,IoT}$ is inversely correlated to $C_{TR}$. This means a higher implementation cost ($C_{TR}$) will lower the upper limit to the cost of a single IoT-RTI. When $c_p = 10$, $C_{TR} = 200$, we have $c_{p,IoT} = 31.5$, when $c_p = 10$, $C_{TR} = 2000$, we have $c_{p,IoT} = 13$. This indicates that the implementation cost will directly affect the transformation benefits of the IoT-RTIs management system. The higher implementation cost, transformation benefits are lower, and the transformation of IoT-RTIs management systems is less worthy. From Table 4 and Figure 12, we also know that when the value of $C_{TR}$ is given, $c_{p,IoT}$ is directly correlated to $c_p$, which means a higher $c_p$ allows a higher upper limit to the cost of a single IoT-RTI with which IoT-RTIs management system adoption is still profitable. When $c_p = 10$, $C_{TR} = 1000$, we have $c_{p,IoT} = 23.3$, when $c_p = 50$, $C_{TR} = 1000$, we have $c_{p,IoT} = 199.7$. This means that if the implementation cost is fixed, the higher value of a single RTI, the greater transformation benefits of the IoT-RTIs management system will be.

**Table 4.** The value of $c_{p,IoT}$ with different $c_p$ and $C_{TR}$.

|  | 10 | 20 | 30 | 40 | 50 |
|---|---|---|---|---|---|
| 200 | 31.5 | 75.6 | 119.7 | 163.8 | 207.9 |
| 400 | 29.4 | 73.5 | 117.6 | 161.7 | 205.8 |
| 600 | 27.4 | 71.5 | 115.6 | 159.7 | 203.8 |
| 800 | 25.3 | 69.4 | 113.5 | 157.6 | 201.7 |
| 1000 | 23.3 | 67.4 | 111.5 | 155.6 | 199.7 |
| 1200 | 21.2 | 65.3 | 109.4 | 153.5 | 197.6 |
| 1400 | 19.2 | 63.3 | 107.4 | 151.5 | 195.6 |
| 1600 | 17.1 | 61.2 | 105.3 | 149.4 | 193.5 |
| 1800 | 15.1 | 59.2 | 103.3 | 147.4 | 191.5 |
| 2000 | 13.0 | 57.1 | 101.2 | 145.3 | 189.4 |

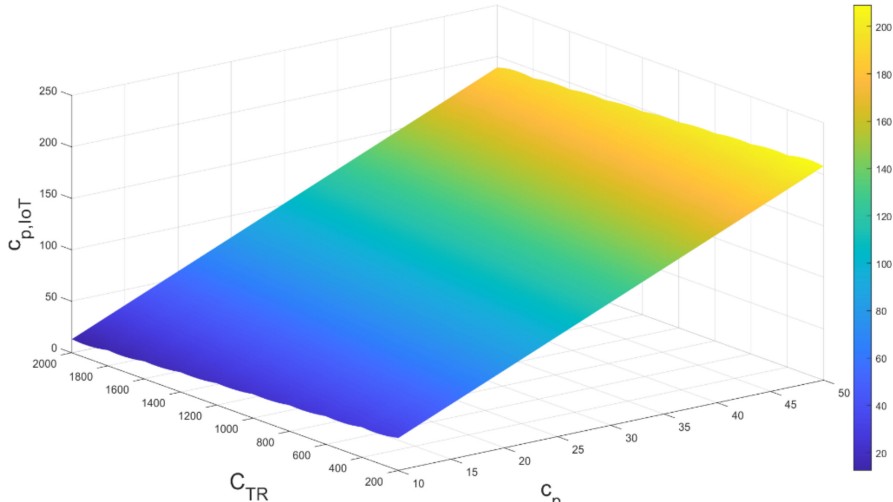

**Figure 12.** Trends of $c_{p,IoT}$ with respect to $c_p$ and $C_{TR}$.

## 4. Discussion

From the above analysis, it can been seen that the return rate of RTIs directly affects the management cost of RTIs in multimodal transport systems. In Sections 2.3.1 and 2.3.3, although the higher return rate of RTIs increases the inventory management cost, the purchase quantity is reduced, and the purchase cost of RTI is often much higher than its inventory management cost in multimodal transport systems. In Section 2.3.2, increasing the return rate also reduces the inventory management cost and purchase of RTIs. This is because RTIs are purchased every $n_2$ cycle, and increasing the return rate can reduce the overall inventory of RTIs on the premise of meeting inventory requirements, thereby reducing their inventory management cost and purchasing cost. Therefore, the higher return rate can effectively reduce the management cost of RTIs in a multimodal transport system. Although the use of IoT technology for RTIs management will increase the software and hardware cost of the system, IoT technology can effectively track the flow status of RTIs in the entire multimodal transport system, record the working status of RTIs (temperature and humidity of the external environment, workload, frequency of use, etc.), which can evaluate and predict the status of RTIs, thereby increasing the return rate and reducing their failure rate and maintenance frequency. Therefore, IoT technology in RTI management can improve its management efficiency and reduce the management costs for the multimodal transport system.

The other goal of the study is to provide organizational management models for companies interested in improving their RTI management. Three schemes are proposed, each suitable for a different situation. For some companies, not having enough RTIs may cause severe damage, and Section 2.3.1 or Section 2.3.2 would be preferable; companies

should consider Section 2.3.1 when they have strong coordination with RTIs suppliers and individual RTIs are more valuable; Section 2.3.2 is better for the opposite scenario, where companies have weak coordination with RTIs suppliers, and individual RTIs are not so valuable; Section 2.3.3 can be adopted by companies for whom not having enough RTIs is not a source of severe damage. Cost models are also established based on the schemes and can be used in cost assessment. We have also analyzed the impact of the IoT-RTIs management system adoption. For companies that have already implemented the above schemes, decision models can be used to determine the conditions under which the use of the IoT-RTIs management system is economical.

## 5. Conclusions

The lack of effective monitoring methods for RTIs has gradually become one of the key problems restricting the sustainable development of multimodal transport systems, and IoT technology is regarded as one of the most important methods to solve this issue. However, the IoT-enabled RTI management system has not been widely accepted by multimodal transport service providers due to the lack of an effective cost decision model to assist enterprises in the intelligent transformation of the RTIs management system. To address these problems, through extensive field studies in collaborative logistics service providers in multimodal transport systems, this research first presents three typical schemes of RTIs management. Then, the cost–benefit analyses of these three schemes were conducted while the decision models on whether to adopt IoT technologies were built. Finally, based on the decision models, the main factors affecting the application of IoT-RTIs management systems were studied by numerical analysis, based on which several managerial implications are presented. These results can serve as a theoretical basis for enterprises interested in finding out whether IoT technology should be used in RTIs management.

There are also some limitations. On the one hand, it assumes a return rate of $0 \leq \alpha \leq 1$, when in practice, there may be RTIs left over from the previous cycle remaining in the IC, which results in a return rate $\alpha > 1$. However, the complexity of the problem can be tremendously increased when $\alpha > 1$ (see Gerchak et al., 1988). To keep the models from becoming too complex, we have made the assumption that $\alpha \leq 1$. In future studies, we may broaden the return rate range to $\alpha \geq 0$ to make the models more universally applicable. On the other hand, we assume the RTIs have fixed and known damage and repairable rates when both parameters are dependent on factors including operators, transport workers, and environments, and variable in reality. In future studies, we may replace them with random variables.

**Author Contributions:** Conceptualization, Y.Z. and X.K.; methodology, X.K.; software, S.Z.; validation, Y.Z. and H.L.; investigation, Y.Z.; resources, H.L.; data curation, L.Q.; writing—original draft preparation, X.K. and Y.Z.; writing—review and editing, Y.Z.; visualization, H.L.; supervision, L.Q.; project administration, X.K.; funding acquisition, X.K., S.Z. and H.L. All authors have read and agreed to the published version of the manuscript.

**Funding:** This work was funded by the Key Scientific Research Project of the University of Henan Province of China (No. 21A460027, No. 20A630036), the Science and Technology Research Project of Henan Province (No. 212102310059), and the General Project of Humanities and Social Sciences of the Ministry of Education of China (No. 21YJC630077).

**Institutional Review Board Statement:** Not applicable.

**Informed Consent Statement:** Not applicable.

**Data Availability Statement:** Not applicable.

**Conflicts of Interest:** The authors declare no conflict of interest.

## Nomenclature

**Notation**

| | |
|---|---|
| $\alpha$ | fraction of returned RTIs after usage |
| $\beta$ | fraction of usable RTIs from returned RTIs, where $0 \leq \beta \leq 1$ |
| $\gamma$ | fraction of RTIs that do not need to be repaired, where $0 \leq \gamma \leq 1$ |
| $c_i$ | variable inspection cost per inspected RTI |
| $c_p$ | cost of purchasing a new RTI |
| $c_r$ | variable repair cost per repaired RTI |
| $c_{p,N}$ | cost of purchasing a common RTI |
| $c_{p,IoT}$ | cost of purchasing an IoT-RTI |
| d | demand rate in units per unit time |
| $h_p$ | cost of keeping an RTI in PF, per RTI per unit of time |
| $h_r$ | cost of keeping a repairable RTI in inventory, per RTI per unit of time |
| $h_u$ | cost of keeping a used RTI in inventory, per RTI per unit of time |
| $h_w$ | cost of keeping an RTI in WC, per RTI per unit of time |
| n | delivery frequency of RTIs from WC to PF in each cycle |
| $n_2$ | procurement cycle of RTIs in Scheme 2 |
| $n_3$ | procurement cycle of RTIs in Scheme 3 |
| $C_I$ | fixed inspection cost per cycle |
| $C_P$ | fixed ordering cost per cycle |
| $C_R$ | fixed repair cost per cycle |
| $C_{TR}$ | implementation cost of the IoT- RTIs management system per cycle |
| T | cycle time |

**Abbreviation**

| | |
|---|---|
| $ETC_1$ | expected total cost of Scheme 1 |
| $ETC_2$ | expected total cost of Scheme 2 |
| $ETC_3$ | expected total cost of Scheme 3 |
| $FC_1$ | fixed cost of Scheme 1 |
| $FC_2$ | fixed cost of Scheme 2 |
| $FC_3$ | fixed cost of Scheme 3 |
| $HC_1$ | inventory holding cost of Scheme 1 |
| $HC_2$ | inventory holding cost of Scheme 2 |
| $HC_3$ | inventory holding cost of Scheme 3 |
| IC | inventory center of RTIs |
| IRC | inventory center of repairable RTIs |
| PC | penalty cost |
| DW | destination warehouse |
| $TC_1$ | total cost of Scheme 1 |
| $TC_2$ | total cost of Scheme 2 |
| $TC_3$ | total cost of Scheme 3 |
| $VC_1$ | variable cost of Scheme 1 |
| $VC_2$ | variable cost of Scheme 2 |
| $VC_3$ | variable cost of Scheme 3 |
| WC | warehouse center |

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
