# Peer review of "IoT-Enabled Sustainable and Cost-Efficient Returnable Transport Management Strategies in Multimodal Transport Systems"

_sustainability, doi:10.3390/su141811668_

Round 1
Reviewer 1 Report
The article examines the conditions for implementing IoT elements in the flow management of various types of Returnable Transport Items (RTI). The authors analyzed three RTI economy models and demonstrated the profitability conditions for their use with the assumed parameters (including stochastic ones).
The article was written correctly and in a good style. The structure of the paper is well worked out, although it would be advantageous to separate the Literature Review.
The content of the article fits the Sustainability journal. However, I encourage the authors to refer to the concept of sustainability in their summary to reinforce this message.
The mathematical formalism used in the article is well thought out and concisely presents the modeled problem. In this regard, I have no significant comments. It is very advantageous to use a table of indexes and markings. Once entered, the index or designation should have the same form throughout the text (italics, bold, font).
The article is an example of good analysis and does not require significant changes. The language of the article is good. However, I suggest the following additions:
1 / The issue of IoT-RTIs system configuration (in terms of sensors, e.g., RFID, hardware, and the management system and its availability for users) has been treated in too little detail, which is related to the implementation costs and maintenance. This can be essential with less expensive RTIs such as pallets.
2 / The authors set the limit of profitability, which is essentially a good approach, but discussing the underlying AoT solutions used in different RTIs and the cost of their maintenance can give a better picture of the cost-effectiveness of the technology.
3 / The authors assume mainly the use of the normal distribution. How is it justified? Should not such an analysis consider a less favorable distribution, e.g., Poisson? (concerns point 3.1)
4 / The authors state that the return rate of RTIs directly affects the management cost of RTIs in multimodal transport systems, which is an expected conclusion. Still, the reference to multimodal transport systems is unclear. The mentioned multimodality is not sufficiently well discussed in the article (what is the definition of multimodality according to the authors?)
5 / The authors state that IoT technology can effectively track the flow status of RTIs in the entire multimodal transport system, but this is a very general statement. Measuring parameters such as humidity and temperature are important, but from the point of view of the load, not necessarily the RTI itself.
Reviewer 2 Report
The paper presents three schemes for Returnable Transport Items management. The text presents some typos and some grammar issues, such as present(s), describe(s), establish(es), analyze(s) (l.16-19), etc. An English native speaker should revise the paper. Separate the sentences with a space at: delays[20] (l.71). The acronyms description from IC, IRC, PC, DW, WC, etc., should be provided. They are in a table, but they should also appear in the text after the first appearance.
The affirmative, beginning in line 36 and ending in line 44, is not referenced. The paragraph beginning in line 65 is extensive. There is a paragraph of just two lines (l.97-97). It should be avoided. The use of words (adjectives) is not recommended: impossible, highly, difficult, etc. Different readers could be different perceptions. They should be able to conclude if something is difficult or not by themselves, for example.
The paper is not properly motivated. Even if there is rare research in RTI management, they must be cited and described, briefly at least. The paper's objective (in the introduction) is unclear, too: how did the authors construct the RTI schemes? What are the paper contributions (it must be written)?
The related works in the introduction describe several works in RTI, but none clearly describes the main challenges this paper is facing. The authors do not provide any comparison to guarantee the use of IoT in RTI.
The authors could cleverly present the nomenclature or add it, at least, to a table. The procurement Schemes have no references. Are they authors' contributions? I suppose so because it is highlighted in the abstract, but it is not clear in the text. Why assume (1-ab)d in Scheme 2? Assuming some values could simplify the approach and make it impossible to use in real cases.
The paper has many formulas. Did the authors create each one of them? For example, formula (1) consists of the fixed and variable costs model. If existing approaches and formulas inspired the authors, they should be referenced.
Does a traditional material flow follow the IC-IRC-Repair-WC-DW-Inspection-IC path? How and why did the authors construct the material flow in that way (Figures 1 to 3)?
The authors do not describe what they mean by IoT-RTI. What kind of technology is used for tracking RTI? Is it being considered middleware and servers infrastructure in costs? Because an RFID tag is a unitary device. But if it transmits information to a centralized server, this cost depends on how many tags the enterprise uses and should be diluted.
Reviewer 3 Report
Please check the attachment.

Round 2
Reviewer 2 Report
The authors provided all my suggestions.
Reviewer 3 Report
All my concerns have been addressed satisfactorily. Thus, this paper can be acceptable.